# Magnitude of client satisfaction and its associated factors with outpatient pharmacy service at Dubti General Hospital, Afar, North East Ethiopia: A cross sectional study

Anwar Brhan Gidey[1], Taklo Simeneh Yazie[2]*, Tegegne Bogale[3], Tesfaye Molla Gulente[4]

1 Quantification and Market Shaping Team Leader, Ethiopian Public Health Institute Focal Person, Samara, Ethiopia, 2 Pharmacology and Toxicology Unit, Department of Pharmacy, College of Health Sciences, Debre Tabor University, Debre Tabor, Amhara, Ethiopia, 3 Department of Pharmacy, Faculty of Medical and Health Sciences, Samara University, Samara, Ethiopia, 4 Coordinator, Health Supply Chain Management, Curative and Rehabilitative Department, Afar National Regional State Health Bureau, Samara, Afar, Ethiopia

* taklosimeneh23@gmail.com

**Data Availability Statement:** All relevant data are within the manuscript and its supporting information file.

## Abstract

### Introduction

In Ethiopia the pharmacy service has had several gaps among these were low patient satisfaction, and poor availability of essential pharmaceuticals. In addition, previous studies showed variation in magnitude of client satisfaction, and there is no previous study in the study area. Therefore, the aim of the study was to determine client satisfaction with outpatient pharmacy service and associated factors among adult clients at Dubti General Hospital in Afar, Ethiopia.

### Methods

A hospital based cross sectional study design was employed from February 1 to March 30, 2020 at Dubti General Hospital. Participants were selected by systematic random sampling method. Bivariate and multivariate binary logistic regression was computed to assess statistical association between the outcome variable, and independent variables. AOR with 95% CI were used to show statistical Significance at P <0.05.

### Results

The overall satisfaction towards outpatient pharmacy service was 165(40.5%). Regarding associated factors, service payment insured through their workplace was positively associated with satisfaction (AOR = 3.178, 95% CI: 1.294–7.80) where as availability of some medications (AOR = 0.393, 95% CI: 0.208–0.741), unfair medication cost (AOR = 0.613, 95% CI: 0.607–0.910), and lack of organized pharmacy work flow (AOR = 0.105, 95% CI: 0.049–0.221) were negatively associated with clients' satisfaction.

**Funding:** The author(s) received no specific funding for this work.

**Competing interests:** The authors have declared that no competing interests exist.

## Conclusion

The clients' satisfaction in this study is low that warrants immediate corrective measures. Corrective measures should be taken based on identified gaps such as improving drug availability, pharmacy work flow, and cost of medications.

## Introduction

Quality of healthcare encompasses the structure, the process, and the outcome of the healthcare [1]. Although providing the highest quality of healthcare is the mission of all healthcare institutions across the world, still poor quality of healthcare is the major problem worldwide. Poor quality of healthcare is now a bigger barrier to reducing mortality than insufficient access, and 60% of deaths from conditions amenable to health care are due to poor quality of healthcare [2]. Patient satisfaction is an indicator of the quality of healthcare, and crucial to evaluate, and to shape the healthcare [3]. Patient's satisfaction is a psychological state that can be defined as a patient's reaction to salient aspect of the context, process, and result of their service experience received from the healthcare institution [4]. Patient satisfaction shows the extent of the gap between the expected service, and the real service received from the patient's judgment [5]. Providing a comprehensive healthcare by giving greater emphasis to patient satisfaction helps improve clinical outcome, and use resources more wisely [6, 7]. Pharmaceutical service is one of the healthcare services provided to patients, and patient satisfaction with the pharmaceutical services helps show the gap in practice, and implementation of the pharmaceutical services [8]. It can affect clinical outcomes, and retention of patients in the healthcare services [9].

Patient satisfaction may vary from country to country as well as region to region due to cultural influences [10, 11], and variation of the healthcare system in terms of number, and competency of healthcare providers [11], and organization of the healthcare institutions [1].

Various studies were conducted to determine the magnitude of patient satisfaction in different countries, and findings of these studies showed considerable percent of patient dissatisfaction [12–17]. In Ethiopia, limited studies were conducted to assess the prevalence of patient satisfaction, and associated factors with pharmacy services. Moreover, the finding of these limited studies was varied ranging from 46.19–59.4% [15–17]. Different factors can affect patient satisfaction, and sex, age, occupation, education level, payment status, availability of drugs, and convenience of waiting area are among identified factors [14–17]. In the study area, there is no previous study to address the magnitude of patient satisfaction, and associated factors at outpatient pharmacy services. Therefore, this study was planned to determine the magnitude of patient satisfaction, and associated factors with outpatient pharmacy services at Dubti General Hospital, Afar, Ethiopia.

## Methods

### Study setting, design, and period

Afar national regional state is one of the national regional states of Ethiopia. It is found in the Northeast Ethiopia, and bordered by Eritrea in the Northeast, and by Djibouti in the East. Afar national regional state has five zones, and its capital city is Samara where it is located 600 kilometers away from Addis Ababa in the northeast. According to the 2017 projected population of Ethiopia, the region has a total population of 1.812 million, and from the total, males, and females account 54.69%, and 45.31%, respectively [18]. The study area is Dubti General

Hospital which is located 12 kilometers away from Samara. It serves around 994,449 peoples, and has several partitions to facilitate service delivery, and Medical ward, Pediatric ward, Surgical ward, and Gynecology ward are among the partitions. It has Anti retroviral therapy pharmacy, Emergency pharmacy, Inpatient pharmacy, and Outpatient pharmacy. A hospital based cross sectional study was applied in this study, and the study was conducted at Dubti General Hospital from February 1 to March 31, 2020.

## Participants and inclusion criteria

All clients who get services from outpatient pharmacy services were considered as source population and the study population was all adult clients who treated as outpatient, and received services from outpatient pharmacy services from February 1 to March 31, 2020. The inclusion criteria were age equal and above 18 years old, clients who treated as outpatient, and received services from outpatient pharmacy services during the study period, and voluntariness to participate in the study. On the contrary, clients who were treated as inpatient, clients with hearing, and mental impairments, age less than 18 years old, and unwillingness to give consent were the exclusion criteria.

## Sample size and sampling procedure

The sample size of this study was determined by taking the magnitude of satisfaction with outpatient pharmacy services as 50% because there is no previous study done in the Afar national regional state regarding this topic. In addition, 5% marginal error, 1.96 Z ($\alpha$1/2), and 95% confidence interval was considered in calculation of the sample size. Based on the above information, and applying the single proportion formula, the sample size became 384. The final sample size was 422 including 10% non response rate.

A systematic random sampling technique was used to include participants in the study. The study area, Dubti General Hospital was selected purposely because of it has high patient flow, and more than 50% of clients in the region were served in Dubti General Hospital. A total of 12,210 clients registered in outpatient pharmacy services in 3 months, and average number of clients who visited monthly the outpatient pharmacy services became 4070. The sampling interval was established by dividing the average monthly number of clients that visited the outpatient pharmacy services with the total sample size that is 4070/422 = 10. Therefore, the sampling interval is 10, and the first client was selected by drawing a number 1 to 10. Then, the data collection was continued by every 10th until the total sample achieved.

## Study variables

Client satisfaction was the outcome variable of the study whereas age, sex, marital status, educational level, occupation, residence, monthly income, pharmacy setting, medication availability, and cost, and convenience of dispensing encounter were the independent variables.

## Data collection

Structured interviewer administered questionnaire were adapted after reviewing different literatures [15, 17]. The questionnaire has two sections; the first contains questions about socio-demographic characteristics, the second contains questions about pharmacy setting, items, and services provided by pharmacy professionals. The questionnaire was prepared in English, and then translated to Amharic, and to local language Afarigna. To check its consistency the questionnaire was back retranslated to English, and Final version of the questionnaire in Afarigna, and Amharic was used to collect the data. The data were collected by 3 diploma holder

health professional that are proficient in writing, and speaking both Amharic, and the local language Afarigna who were trained on the objective of the study, techniques of interviewing, how to approach clients, and keeping confidentiality with close supervision by a degree holder pharmacist, and principal investigator. Exit interview was conducted in the Green area far from the dispensary after clients received pharmaceutical services to avoid interviews being overheard by pharmacy staff.

## Data quality control

Data quality was ensured by pretest on 10% of the total sample size at Logiya Hospital, by giving training to data collectors about the purpose, and methodology of the study. In addition, the supervisor checked the data collectors, and the collected data daily for completeness, and accuracy. The principal investigator supervised both the supervisors, and data collectors to maintain the completeness, and accuracy of the collected data. Moreover, the reliability of the questionnaire was checked, and found to be very good (Cronbach's alpha = 0.70).

## Statistical methods

The level of client satisfaction was rated out of five. From the five point likert scale, 1, 2, 3, 4, and 5 stands for strongly disagree, disagree, neutral, agree, and strongly agree, respectively. For descriptive interpretation, the five scales were converted into a three scale format by combining very satisfied and satisfied as satisfaction, and dissatisfied and very dissatisfied as dissatisfaction.

The normal distribution of data was checked by using Q-Q plot box, and the result showed that the plotted data closed to the straight line, which is a characteristic of normally distributed data. In addition, in checking of the mean outlier status, the cock distance result showed minimum 0.00, and maximum 0.26 which is below 1, indicates no outlier with mean. The mean level of clients' satisfaction was calculated by averaging their ratings for the items measuring satisfaction, and those who scored greater than the overall mean value were considered to be satisfied, and those who scored less than or equal to the overall mean value were considered to be dissatisfied with outpatient pharmacy services.

The multicolinarity effects between the independent variables were checked by using variance inflation factors (VIF), and VIF was found to be below 2 which indicate that there are no multicolinarity effects.

Data from all the questionnaires were coded, entered into and analyzed by using SPSS package version 21. Frequencies, mean with standard deviation, and cross tabulation were used to describe the data. Associations between independent variables, and dependent variable were analyzed using binary logistic regression. Factors with p-value of <0.2 in the bivariate logistic regression were included in the multivariate logistic regression analysis. The magnitude of the association was measured by using odds ratios, and 95% confidence interval (CI), and a p-value of <0.05 was considered statistically significant.

## Ethical consideration

The study was carried out after getting ethical approval from Samara University, College of Medical and Health Sciences, Ethical Review Committee (ERC 006/2020). After reviewing the study protocol, the Ethical Review Committee accepted and approved the request of verbal consent as the study does not have any harm to participants. Authors chose verbal consent rather than written consent Authors chose verbal consent rather than written consent due to previous experiences that participants prefer verbal consent as they think, it is preferred to assure their confidentiality. Permission letter was received from Samara University, College of

Medical and Health Sciences to facilitate the study, and given to Dubti General Hospital. Before data collection, verbal consent was obtained from all participants after they were told about the objective, and the methodology of the study. Personal identifiers were anonymized to keep the confidentiality of the participants. This study was conducted based on the principles of Declaration of Helsinki.

## Operational definitions

✓ Client: Patient who received outpatient pharmacy services during study period.

✓ Satisfaction: Clients who scored above the overall mean towards outpatient pharmacy service.

✓ Dissatisfaction: Clients who scored below or equal to the overall mean towards outpatient pharmacy service.

## Results

### Socio-demographic characteristics of study participants

From a total sample size of 422, 407 clients were included in the final analysis with a response rate of 96%, and 15 participants refused to participate in the study. The study participants were predominantly male which comprised of 218 (53.6%). Regarding to marital status, and age groups, being married, and age group of 29–39 years old comprised 224 (55.0%), and 160 (39.5%), respectively. Majority of clients were employed in governmental institutions (100, 24.6%), and lived in urban areas (326, 80.1%) (Table 1).

**Table 1. Socio-demographic characteristics of the study participants (N = 407).**

| Variable | Category | Frequency | Percent | Overall satisfaction | |
|---|---|---|---|---|---|
| | | | | Satisfied (%) | Dissatisfied (%) |
| **Sex** | Male | 218 | 53.6 | 85(39) | 133(61) |
| | Female | 189 | 46.4 | 80(42.7) | 109(57.3) |
| **Age in years** | 18–28 | 144 | 35.4 | 56(38.9) | 88(61.1) |
| | 29–39 | 160 | 39.5 | 71(44.4) | 89(55.6) |
| | 40–49 | 82 | 20.1 | 30(36.6) | 52 (63.4) |
| | ≥50 | 21 | 5.2 | 8(38.1) | 13(61.9) |
| **Marital status** | Married | 224 | 55.0 | 91(40.6) | 133(59.4) |
| | Divorced | 34 | 8.4 | 11(40.7) | 16(59.3) |
| | Single | 149 | 36.6 | 58(38.9) | 91(61.1) |
| **Occupation** | Government | 100 | 24.6 | 31(31.0) | 69(69.0) |
| | Unemployed | 54 | 13.3 | 24(44.4) | 30(55.6) |
| | Merchant | 63 | 15.5 | 24(38.1) | 39(61.9) |
| | Daily laborers | 60 | 14.7 | 24(40.0) | 36(60.0) |
| | Student | 88 | 21.6 | 36(40.9) | 52(59.1) |
| | Pastoral | 42 | 10.3 | 26(61.9) | 16(38.1) |
| **Current residence** | Urban | 326 | 80.1 | 118(36.2) | 208(63.8) |
| | Rural | 81 | 19.9 | 47(58.0) | 34(42.0) |
| **Payment Status** | In cash | 344 | 84.5 | 153(44.5) | 191(55.5) |
| | Credit | 63 | 15.5 | 12(19.0) | 51(81.0) |

## Clients' response towards pharmacy setting, and availability of pharmaceutical items at outpatient pharmacy

More than 90% of participants agreed with the appropriateness of the location of the outpatient pharmacy. In addition, over 82% of clients agreed on the convenience of waiting area, and dispensary and counter. Concerning the cleanness of the outpatient pharmacy unit, 357 (88.1%) clients agreed on the cleanness of the outpatient pharmacy unit. Among all clients, only 153 (37.6%) clients got their all prescribed drugs from the outpatient pharmacy unit.

## Satisfaction scores of patients towards outpatient pharmacy services

More than two-third of the sampled clients were satisfied by the organized work flow of the outpatient pharmacy services. Above 81% of clients had satisfaction on how to take medications, and 314 (77.1%) clients were found to have satisfaction on the label readability and understandability. Moreover, more proportion of clients had satisfaction on service waiting time, respect and politeness of pharmacists, and willingness to answer questions. However, lower proportion of clients had satisfaction with regard to cost of available prescribed drugs, counseling on possible side effects of drugs, drug interaction, counseling time, and storage condition of drugs (Table 2).

## Client level of satisfaction towards outpatient pharmacy services

On the basis of the five-point likert scale, the mean score of satisfaction was 3.12. The cut off point for determination of clients' satisfaction was the overall mean score that is 3.12. Based on this cut off point, 165 (40.5%) clients were found to have satisfaction with outpatient pharmacy services by scoring greater than the mean.

**Table 2. The satisfaction status of clients at Dubti General Hospital, Afar, Ethiopia, 2020.**

| Variables | Satisfied | Neutral | Dissatisfied |
|---|---|---|---|
| | Frequency, N (%) | Frequency, N (%) | Frequency, N (%) |
| The prescribed drugs are avail in the pharmacy | 153(37.6%) | 231(56.8%) | 23(5.6%) |
| Are you satisfied with the cost of the medication | 125(30.7%) | 3(0.7%) | 279(68.6%) |
| Location of the pharmacy is comfortable | 381(90.3%) | 15(6.1%) | 10(3.6%) |
| Pharmacy location related with other service | 298(74.2%) | 19(4.5%) | 90(21.3%) |
| Waiting area is clean and comfortable | 335(79.4%) | 36(11.8%) | 36(8.5%) |
| The pharmacy have organized work flow | 303(71.8%) | 46(10.9%) | 58(17.30%) |
| Cleanness of Outpatient pharmacy | 335(82.9%) | 51(12.1%) | 21(5.0%) |
| Convenience of dispensing area and counter | 357(88.1%) | 13(3.1%) | 37(8.8%) |
| Service equally delivered | 316(77.6%) | 19(4.7%) | 72(17.7%) |
| Willing to answer question | 322(73.2%) | 16(3.9%) | 69(16.9%) |
| Easy and understandable language | 314(77.1%) | 12(3%) | 81(19.9%) |
| Respect and politeness | 274(67.3%) | 31(7.6%) | 102(25.1%) |
| Service waiting time | 216(53.1%) | 22(5.4%) | 169(41.5%) |
| Medication administration | 333(81.9%) | 6(1.4%) | 68(15.7%) |
| Possible side effects | 130(31.9%) | 8(2.0%) | 269(66.1%) |
| Medication interactions | 65(15.9%) | 2(.5%) | 340(83.6%) |
| Storage of medicines | 125(30.8%) | 1(.4%) | 281(69.0%) |
| Labeling on dispensed drugs | 263(64.6%) | 7(1.7%) | 113(33.7%) |
| Spends enough counseling time | 153(37.0%) | 9(2.2%) | 245(60.2%) |

**Table 3. Multivariable logistic regression analysis of factors at Dupti General Hospital, Afar, Ethiopia, 2020.**

| Variables | | Satisfaction status, N (%) | | COR (95% CI) | AOR (95% CI) | P-value |
|---|---|---|---|---|---|---|
| | | Satisfied | Dissatisfied | | | |
| Occupation | Governmental | 31(31.0) | 69(69.0) | 1 | 1 | |
| | Unemployed | 24(44.4) | 30(55.6) | 0.562(0.283,1.113) | 0.759(0.233,2.47) | 0.647 |
| | Student | 36(40.9) | 52(59.1) | 0.649(0.356,1.183) | 5.347(0.636,44.98) | 0.123 |
| | Others | 26(61.9) | 16(38.1) | 0.276(0.130,0.587) * | 0.243(0.060,0.984) | 0.047 |
| Current residence | Urban | 118(36.2) | 208(63.8) | 1 | 1 | |
| | Rural | 47(58.0) | 34(42.0) | 0.410(0.250,0.674)* | 0.452(0.190,1.078) | 0.073 |
| Payment status | Cash | 153(44.5) | 191(55.5) | 1 | 1 | |
| | Credit | 12(19.0) | 51(81.0) | 0.294(0.151,0.57) * | 3.178(1.294,7.80) | 0.012 |
| Monthly income | Low | 23(56.1) | 18(43.9) | 1 | 1 | |
| | Average | 23(33.8) | 45(66.2) | 2.500(1.128,5.53)* | 0.514(0.050,5.30) | 0.577 |
| | Above average | 21(32.3) | 44(67.7) | 2.677(1.195,5.99) * | 1.440(0.455,4.55) | 0.535 |
| | High | 33(37.9) | 54(62.1) | 2.091(0.984,4.44) | 1.802(0.663, 4.89). | 0.248 |
| | No income | 65(44.5) | 81(55.5) | 1.592(0.792,3.20) | 0.348(0.376,4.829 | 0.100 |
| Medication availability | All | 40(26.1) | 113(45.9) | 1 | | |
| | Some | 115(49.8) | 116(50.2) | 0.350(0.219,0.560) | 0.393(0.208,0.741) | 0.003 |
| | No | 5(21.0) | 18(79.0) | 0.360(0.20, 2.642) * | 0.460(0.218,1.014) | 0.091 |
| Fairness of medication cost | Yes | 45(36.0) | 80(64.0) | 1 | 1 | |
| | No | 80(28.1) | 160(71.3) | 0.756(0.489,1.169)* | 0.613(0.607,0.910) | 0.024 |
| Pharmacy location | Yes | 140(37.8) | 230(62.2) | 1 | 1 | |
| | Neutral | 4(28.6) | 10(71.4) | 0.166(0.046,0.605) | 0.105(0.022,0.511) | 0.057 |
| | No | 7(30.4) | 16(69.6%) | 0.391(0.165,0.928) * | 0.362(0.124,1.057) | 0.063 |
| Pharmacy work flow | Yes | 90(30.8) | 202(69.2) | 1 | 1 | |
| | Neutral | 21(46.7) | 24(53.3) | 0.509(0.270,0.962)* | 0.598(0.264,1.35) | 0.204 |
| | No | 54(77.1) | 16(22.9) | 0.132(.072,0.243) * | 0.105(0.049,0.221) | 0.0001 |

Note

* stands for P <0.2 in bivariate logistic regression; others include pastoral, daily laborer, and merchant.

## Factors associated with patient satisfaction towards outpatient pharmacy services

In the bivariate logistic regression analysis, occupation, residence, payment status, monthly income, drug availability, medication cost, pharmacy location, and workflow were showed a significant association with the outcome variable. Occupation, payment status, drug availability, medication cost, and pharmacy work flow retained their significant statistical association with the outcome variable in multivariate logistic regression. Clients missed either one or more prescribed medication were less likely to be satisfied (AOR = 0.393, 95% CI: 0.208–0.741, P = 0.003) compared to those patients who have gotten all prescribed drugs. Moreover, clients who thought that cost of prescribed medications was unaffordable, were less likely to be satisfied (AOR = 0.613, 95% CI: 0.607–0.910, P = 0.024) compared to clients who thought that cost of prescribed medications was affordable (Table 3).

## Discussion

Client satisfaction with pharmaceutical services provided has been recognized as a measure of outcome of the healthcare, and helps improve the quality of the pharmacy services [19]. Therefore, the present study determined the magnitude of client satisfaction and associated factors

with outpatient pharmacy services. In this study, clients' satisfaction with outpatient pharmacy services was (40.5%) which is lower than the findings of other studies in Tanzania (46%) [12], South Korea (74.6%) [13], Mizan Tepi University Teaching Hospital (52.6%) [14], Brazil (58.4%) [15], Tikur Anbessa Specialized Hospital (51.6%) [20], Eastern Ethiopia Public Hospitals (46.19%) [16], and Dessie Referral Hospital and Borumeda Hospital (59.4%) [17]. The possible reason for the discrepancy of the findings might be the difference in weather condition, awareness of clients about pharmacy services, and demographic characteristics. The weather condition in Afar is hot where clients might not feel comfortable, and higher waiting time for the service can more disappoint the client compared to clients in favorable weather condition. In addition, pharmacists in harsh environment might not have interest as equal as pharmacists work in favorable weather condition to serve the clients.

In the present study, the maximum satisfaction was achieved with appropriateness of pharmacy location where as in other studies in Ethiopia, the maximum client satisfaction was obtained with information provided about medication administration, and the politeness and interest of pharmacists in South Wollo Hospitals [17], and Tikur Anbessa Specialized Hospital [20], respectively. In the current study, the lowest satisfaction was observed in fairness of cost of medications, counseling time, information provided about medication storage conditions, medication interactions, and side effects. In these areas, the outpatient pharmacy services should be improved, and gotten more emphasis to improve the overall quality of pharmacy services.

Different factors were associated with clients' satisfaction in this study. Occupation, payment for services, work flow of outpatient pharmacy, medication availability, and medication cost were significantly associated with satisfaction. From these factors, getting outpatient pharmacy service with credit payment status was positively associated with satisfaction whereas the remaining mentioned factors were negatively associated with clients' satisfaction. Regarding to payment status, the finding of this study is in line with the finding of the study conducted in South Wollo Hospitals where clients who were covered their health cost through company were more satisfied than clients who paid out of pocket [17]. Similarly, the finding of the study supported the current study in that free fee paid or lower health cost increased the satisfaction of clients towards outpatient pharmacy services [13].

In this study clients' response of not convenient pharmacy work flow, and somewhat fair was negatively associated with clients' satisfaction, which is supported by the findings of other studies done in Tikur Anbessa Specialized Hospital, and in South Wollo Public Hospitals where pharmacy setting was associated with satisfaction [17, 20]. Availability of some, and none of all prescribed drugs were negatively associated with clients' satisfaction which is in line with the findings of other studies [16, 17, 21]. Cost of medication was an important factor of this study, and clients' response of medication cost is not fair was negatively associated with clients' satisfaction. This finding was supported by the finding of the study done in South Korea where free fee paid or lower health cost increased satisfaction [13].

In contrast to the present study, different studies showed that age, gender, marital status, level of education were associated with clients' satisfaction [15, 17, 22]. The difference might be in part due to variation in study period, and sample size of the studies. Regarding to occupation in this study, daily laborers, pastoral, and merchants together were less likely to be satisfied with pharmacy services compared to clients who were employed in governmental institutions. The reason for this difference needs further study.

Our study has strength and limitation. This study was the first study which addressed the magnitude of clients' satisfaction and associated factors at Dubti General Hospital in Afar, Ethiopia, and this could be taken as strength of the study. On the other hand, the following could be the limitation of our study. Being cross sectional that might not show the actual

clients satisfaction over a period of time. In addition, difference in awareness, and experience of clients with pharmacy services were not assessed, and these might affect the outcome of the study.

## Conclusion

Measuring the level of patient satisfaction using various dimensions of services helps to predict the gap between patient needs and patient satisfaction regarding pharmaceutical care services. The current cross-sectional study reported that majority of the participants was dissatisfied with outpatient hospital pharmacy services. Implementing of auditable pharmaceutical transaction system (APTS) influences the clients satisfaction towards outpatients pharmacy service, and the study also revealed that medication guidance is an area of attention to improve overall patient care process.

## Supporting information

**S1 File. Data collection tool (questionnaire).**
(DOCX)

## Acknowledgments

Authors acknowledged Mr. Yemane Gebre Mariam for his valuable comments, and overall advisor ship to complete this study. In addition, authors gave thank to Samara University, Dubti General Hospital, and data collectors for their support towards completion of the study.

## Author Contributions

**Conceptualization:** Anwar Brhan Gidey, Taklo Simeneh Yazie, Tegegne Bogale, Tesfaye Molla Gulente.

**Data curation:** Anwar Brhan Gidey, Taklo Simeneh Yazie, Tegegne Bogale, Tesfaye Molla Gulente.

**Formal analysis:** Anwar Brhan Gidey, Taklo Simeneh Yazie, Tegegne Bogale, Tesfaye Molla Gulente.

**Investigation:** Anwar Brhan Gidey, Taklo Simeneh Yazie, Tegegne Bogale, Tesfaye Molla Gulente.

**Methodology:** Anwar Brhan Gidey, Taklo Simeneh Yazie, Tegegne Bogale, Tesfaye Molla Gulente.

**Project administration:** Anwar Brhan Gidey, Taklo Simeneh Yazie, Tegegne Bogale, Tesfaye Molla Gulente.

**Resources:** Anwar Brhan Gidey, Taklo Simeneh Yazie, Tegegne Bogale, Tesfaye Molla Gulente.

**Software:** Anwar Brhan Gidey, Taklo Simeneh Yazie, Tegegne Bogale, Tesfaye Molla Gulente.

**Supervision:** Anwar Brhan Gidey, Taklo Simeneh Yazie, Tegegne Bogale, Tesfaye Molla Gulente.

**Validation:** Anwar Brhan Gidey, Taklo Simeneh Yazie, Tegegne Bogale, Tesfaye Molla Gulente.

**Visualization:** Anwar Brhan Gidey, Taklo Simeneh Yazie, Tegegne Bogale, Tesfaye Molla Gulente.

**Writing – original draft:** Anwar Brhan Gidey, Taklo Simeneh Yazie, Tegegne Bogale, Tesfaye Molla Gulente.

**Writing – review & editing:** Anwar Brhan Gidey, Taklo Simeneh Yazie, Tegegne Bogale, Tesfaye Molla Gulente.

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
