## [Decision Letter · Decision Letter 0]

20 Oct 2021

PONE-D-21-25324Magnitude of client satisfaction and its associated factors with outpatient pharmacy service at Dubti General Hospital, Afar, North East Ethiopia: A cross sectional studyPLOS ONE

Dear Dr. taklo simeneh yazie%,

Thank you for submitting your manuscript to PLOS ONE. After careful consideration, we feel that it has merit but does not fully meet PLOS ONE’s publication criteria as it currently stands. Therefore, we invite you to submit a revised version of the manuscript that addresses the points raised during the review process.

We look forward to receiving your revised manuscript.

Kind regards,

Bin Su, Ph.D.

Academic Editor

PLOS ONE

Journal Requirements:

2. In the ethics statement in the Methods and online submission information, please clarify whether consent was written or verbal.  If verbal, please also specify: 1) whether the ethics committee approved the verbal consent procedure, 2) why written consent could not be obtained, and 3) how verbal consent was recorded. If your study included minors, state whether you obtained consent from parents or guardians. If the need for consent or parental consent was waived by the ethics committee, please include this information

3. Please include additional information regarding the survey or questionnaire used in the study and ensure that you have provided sufficient details that others could replicate the analyses. For instance, if you developed a questionnaire as part of this study and it is not under a copyright more restrictive than CC-BY, please include a copy, in both the original language and English, as Supporting Information

Reviewers' comments:

Reviewer's Responses to Questions

**Comments to the Author**

1. Is the manuscript technically sound, and do the data support the conclusions?

Reviewer #1: Yes

Reviewer #2: Yes

2. Has the statistical analysis been performed appropriately and rigorously? 

Reviewer #1: Yes

Reviewer #2: Yes

3. Have the authors made all data underlying the findings in their manuscript fully available?

Reviewer #1: Yes

Reviewer #2: Yes

4. Is the manuscript presented in an intelligible fashion and written in standard English?

Reviewer #1: Yes

Reviewer #2: Yes

5. Review Comments to the Author

Reviewer #1: The authors performed a investigation about the client satisfaction with outpatient pharmacy service at Dubti General Hospital. Bivariate and multivariate binary logistic regression was computed to assess statistical association. The data was clearly described the relationships between client satisfaction and some variables, like payment status, medication cost and organized pharmacy work flow. Moreover, the manuscript has been also well written. Thus, the manuscript could be publishable.

Reviewer #2: The manuscript by Gidey et al. describes a hospital based cross-sectional study of client satisfaction and its associated factors with outpatient pharmacy service at Dubti General Hospital, Afar, North East Ethiopia. Several other groups worldwide has performed similar studies to identify the critical factors that affect the health care systems. As this is the first study performed in Ethiopia region, the authors did considered many important aspects of the clients in order to improve the healthcare and pharmacy services. In my opinion, the authors may also need to improve the preparation of the manuscript in the following aspects.

1. The population demographics are given from 2007, more recent data would be better.

2. While considering the different variables with respects to clients, pharmacist and pharmacy, the author may also include the skills and knowledge of the pharmacist, and any insurance that covers the client’s payment.

3. Language quality of the manuscript need to be improved in grammatical and spell check.

6. PLOS authors have the option to publish the peer review history of their article (what does this mean?). If published, this will include your full peer review and any attached files.

Reviewer #1: **Yes: **Yaxin Li

Reviewer #2: No

---

## [Author Response · Author response to Decision Letter 0]

26 Oct 2021

Author responses to comments

Journal Requirements:

Author response: Dear Editor, we saw the link you invited me (The PLOS ONE style templates can be found at https://journals.plos.org/plosone/s/file?id=wjVg/PLOSOne_formatting_sample_main_body.pdf and https://journals.plos.org/plosone/s/file?id=ba62/PLOSOne_formatting_sample_title_authors_affiliations.pdf) and I made corrections to the manuscript style to meet the requirement of PLOS ONE.

2. In the ethics statement in the Methods and online submission information, please clarify whether consent was written or verbal. If verbal, please also specify: 1) whether the ethics committee approved the verbal consent procedure, 2) why written consent could not be obtained, and 3) how verbal consent was recorded. If your study included minors, state whether you obtained consent from parents or guardians. If the need for consent or parental consent was waived by the ethics committee, please include this information

Author response: Dear Editor, the consent obtained was verbal informed consent. 1) The ethical Review Committee approved the verbal consent after reviewing the study protocol. 2) Committee accepted and approved the request of verbal consent as the study does not have any harm to participants. Authors chose verbal consent rather than written consent Authors chose verbal consent rather than written consent due to previous experiences that participants prefer verbal consent as they think, it is preferred to assure their confidentiality. 3) Detail of the study’s objective, method of data collection, and ethical concern was explained to potential participants, and then, were requested to participate in the study voluntarily. For volunteer participants, data collectors tick right sign in front of I agree, then continue interviewing the participant. There were no minors in this study, all are adults who can themselves give consent after understanding what were explained about the study.

3. Please include additional information regarding the survey or questionnaire used in the study and ensure that you have provided sufficient details that others could replicate the analyses. For instance, if you developed a questionnaire as part of this study and it is not under a copyright more restrictive than CC-BY, please include a copy, in both the original language and English, as Supporting Information

Author response: Dear Editor, based on your request we included the verbal consent form, and the questionnaire (in English version, Amharic version, and Afarigna version) as supporting information.

Author response: we reviewed references we used, and we replaced reference 2 due to it has some correction issues (Kruk ME, Gage AD, Arsenault C, Jordan K, Leslie HH, Roder-DeWan S, Adeyi O, Barker P, Daelmans B, Doubova SV, English M replaced by Kruk ME, Gage AD, Joseph NT, Danaei G, García-Saisó S, Salomon JA, 2018). 

5. Review Comments to the Author

Reviewer #1: The authors performed a investigation about the client satisfaction with outpatient pharmacy service at Dubti General Hospital. Bivariate and multivariate binary logistic regression was computed to assess statistical association. The data was clearly described the relationships between client satisfaction and some variables, like payment status, medication cost and organized pharmacy work flow. Moreover, the manuscript has been also well written. Thus, the manuscript could be publishable.

Reviewer #2: The manuscript by Gidey et al. describes a hospital based cross-sectional study of client satisfaction and its associated factors with outpatient pharmacy service at Dubti General Hospital, Afar, North East Ethiopia. Several other groups worldwide has performed similar studies to identify the critical factors that affect the health care systems. As this is the first study performed in Ethiopia region, the authors did considered many important aspects of the clients in order to improve the healthcare and pharmacy services. In my opinion, the authors may also need to improve the preparation of the manuscript in the following aspects.

1. The population demographics are given from 2007, more recent data would be better.

Author response: Dear reviewer, I made corrections as per your comment: reference 18 was replaced due to its oldness (CSA, 2007 replaced by Population Projection of Ethiopia, 2017)

 2. While considering the different variables with respects to clients, pharmacist and pharmacy, the author may also include the skills and knowledge of the pharmacist, and any insurance that covers the client’s payment.

Author response: Dear reviewer, from the beginning we considered factors related to pharmacists skill and knowledge (guidance of pharmacists about medications) but they were failed to fulfill the criteria of p<0.2 to be incorporated in to multivariate analysis. Regarding to insurance, we used the term credit to include getting free services as a result of receiving support due to poverty, health insurance, company coverage of individual health care cost. Therefore, health insurance was considered in the analysis of the study.

 3. Language quality of the manuscript need to be improved in grammatical and spell check

Author response: Dear reviewer, as per your comment I revise the manuscript for grammatical and spelling errors. The corrections highlighted in the document entitled “revised manuscript with track changes”

---

## [Editor Report · Decision Letter 1]

3 Nov 2021

Magnitude of client satisfaction and its associated factors with outpatient pharmacy service at Dubti General Hospital, Afar, North East Ethiopia: A cross sectional study

PONE-D-21-25324R1

Dear Dr. %taklo simeneh yazie%,

We’re pleased to inform you that your manuscript has been judged scientifically suitable for publication and will be formally accepted for publication once it meets all outstanding technical requirements.

Kind regards,

Bin Su, Ph.D.

Academic Editor

PLOS ONE
---

## [Editor Report · Acceptance letter]

8 Nov 2021

PONE-D-21-25324R1 

Magnitude of client satisfaction and its associated factors with outpatient pharmacy service at Dubti General Hospital, Afar, North East Ethiopia: A cross sectional study 

Dear Dr. Yazie:

I'm pleased to inform you that your manuscript has been deemed suitable for publication in PLOS ONE. Congratulations! Your manuscript is now with our production department. 

Kind regards, 

on behalf of

Dr. Bin Su 

Academic Editor

PLOS ONE